# Integration of observed and model derived groundwater levels in landslide threshold models in Rwanda

Judith Uwihirwe[1, 2], Markus Hrachowitz[1], Thom Bogaard[1]

[1]Section Water Resources, Department of Water Management, Faculty of Civil Engineering and Geosciences, Delft University of Technology, PO Box 5048, 2600 GA, Delft, The Netherlands
[2]Department of Irrigation and Drainage, School of Agricultural Engineering, College of Agriculture Animal Sciences and Veterinary Medicine, University of Rwanda, PO Box 210, Musanze, Rwanda

*Correspondence to*: Judith Uwihirwe (uwihirwejudith@yahoo.fr)

**Abstract.** Incorporation of specific regional hydrological characteristics in empirical statistical landslide threshold models has considerable potential to improve the quality of landslide predictions towards reliable early warning systems. The objective of this research was to test the value of regional groundwater level information, as a proxy for water storage fluctuations, to improve regional landslide predictions with empirical models based on the concept of threshold levels. Specifically, we investigated: i) the use of a data driven time series approach to model the regional groundwater levels based on short duration monitoring observations; ii) the predictive power of single variable and bilinear threshold landslide prediction models derived from groundwater levels and precipitation. Based on statistical measures of the model fit ($R^2$ and RMSE), the groundwater level dynamics estimated by the transfer function noise time series model are broadly consistent with the observed groundwater levels. The single variable threshold models derived from groundwater levels exhibited the highest landslide prediction power with 82–93 % of true positive alarms despite the quite high rate of false alarms with about 26–38 %. Further combination as bilinear threshold models reduced the rate of false alarms by about 18–28 % at the expense of reduced true alarms by about 9–29 % and thus, being less advantageous than single variable threshold models. In contrast to precipitation based thresholds, relying on threshold models exclusively defined using hydrological variables such as groundwater can lead to improved landslide predictions due to their implicit consideration of long term antecedent conditions until the day of landslide occurrence.

## 1 Introduction

Landslide as well as other natural hazard prediction models are developed with purpose of being implemented into early warning systems (LEWS) (Fathani et al., 2016; Pecoraro et al., 2019; Piciullo et al., 2018). LEWS are defined as tools to monitor the long term hydrological and short term meteorological variations to predict and timely inform about the imminent periods of landslide danger. Most landslide prediction approaches and development of early warning criteria routinely rely on meteorological threshold level concepts which define the typical precipitation characteristics like event rainfall volume, event rainfall intensity and event duration that initiate landslides (e.g. Guzzetti et al., 2008; Brunetti et al., 2010; Rosi et al., 2016; Peruccacci et al., 2017). However, this exclusive reliance on meteorological data is problematic for several reasons. The most

common problem attributed to these meteorological threshold level concepts is the frequent lack of considering pre-event hydrological processes and specific characteristics of the studied region that predispose the slope to near failure (Bogaard and Greco, 2014, 2018; Mostbauer et al., 2018; Peres et al., 2017). These approaches are therefore known to generate high rates of false and missed alarms and thus, reducing the quality of landslide early warning systems. Hydrology, being an important aspect in landslide hazard assessment, is still not sufficiently explored although many landslides are hydrologically caused and meteorologically triggered (Bogaard and Greco, 2018). While landslides are hydrologically caused by elevated pre-event subsurface water storage, they are meteorologically triggered by the input of precipitation and snow melt during a specific event that lead to a further increase in pore water pressure, a decrease in frictional forces between particles that reduces the effective shearing resistance and thus create slope instability (van Beek, 2002; Bishop, 1954; Kuriakose, 2010). According to Bogaard and Greco (2014, 2015), the integration of hydrological processes into large scale models is still incomplete and therefore, limited the application into landslide prediction models. The need for landslide hydrological–meteorological based thresholds was highlighted and further postulated that both false and missed alarms could be significantly reduced if the wetness state is incorporated in landslide prediction models through direct measurements of soil water content or groundwater levels. However, various ways of including such hydrological state information into landslide hydro–meteorological thresholds have been recently attempted. These include the direct use of in situ hydrological data through standard observation networks such as stream flow or local soil moisture observations (e.g. Mirus et al., 2018b; Wicki et al., 2020) but also data from satellite derived hydrological measurement (e.g. Zhuo et al., 2019; Thomas et al., 2019; Marino et al., 2020; van Natijne et al., 2020) as well as hydrological variables estimated from hydrological models (e.g. Ciavolella et al., 2016; Mostbauer et al., 2018; Prenner et al., 2018, 2019; Wang et al., 2019; Zhao et al., 2020). It should be noted that research that incorporates hydrological parameters into landslide prediction models using in situ data is scarce due to absence of long term hydrological monitoring of sufficient spatial and temporal coverage in most regions worldwide. This is in particular true for many African countries, where the underlying problem limiting landslide research is the lack of sufficient local data. Freely available satellite and global hydrological model derived information is also still poorly explored. In Rwanda, many river catchments have been recently equipped with groundwater observation wells, piezometers and river water level gauges. However, frequently, the recorded data are insufficient to build historical time series that match the time period of landslide inventories and that could be incorporated into landslide hydro–meteorological threshold model. Recently, Uwihirwe et al. (2020) published the first empirical landslide hazard assessment relation for Rwanda, which is an important step forward in landslide early warning in that country. The defined precipitation based landslide threshold included the antecedent precipitation conditions as an indirect proxy for hydrological conditions. However, it still suffers from an elevated number of false and missed alarms. Recent research suggests that the number of false alarms can be reduced once the hydrological state information are incorporated in landslide prediction models. Several papers reported significant improvement of landslide forecast quality for early warning system by replacing the antecedent rainfall component with soil moisture data (Mirus et al., 2018b; Mostbauer et al., 2018; Prenner et al., 2018, 2019; Zhuo et al., 2019; Thomas et al., 2019; Wang et al., 2019; Marino et al., 2020; Zhao et al., 2020; van Natijne et al., 2020; Wicki et al., 2020). The need for landslide hydro–meteorological thresholds is therefore widely

acknowledged. However, the functional relationship between hydrological and meteorological conditions potentially linked to landslide initiation is not yet standardized. Traditional precipitation based threshold models commonly used power–law functions between precipitation variables like intensity–duration I–D and event–duration E–D (e.g. Caine, 1980; Guzzetti et al., 2007, 2008; Ma et al., 2015; Hong et al., 2017) using the threshold model line as the best separator for landslide and no landslide conditions sometimes defined based on the experts judgment. More advanced statistical approaches that include the

frequentist, probabilistic and receiver operating characteristics methods have been adopted and replaced the deterministic method. The frequentist method (Brunetti et al., 2010; Melillo et al., 2018; Peruccacci et al., 2017; Piciullo et al., 2018) also defines the threshold line separating landslide from no landslide conditions based on the targeted exceedance probabilities. The probabilistic method (Berti et al., 2012; Robbins, 2016) fundamentally rely on Bayes' prior and marginal probabilities for landslide occurrence. The probabilistic methods are criticized for the biased prior and marginal probabilities due to the

incompleteness of typical landslide inventory data (Berti et al., 2012) while frequentist methods are constrained by their high dependency on a large and well distributed dataset to achieve significant results (Brunetti et al., 2010; Monsieurs et al., 2019). The receiver operating characteristic ROC curve method compares the landslide and no landslide conditions based on the area under the curve AUC while indicating the trade–off between true and false positive rates associated to each level of the tested predictor variable or model. In landslide studies, the ROC approach has been mostly used to evaluate the performance of

landslide prediction models (Hong et al., 2017; Wicki et al., 2020) despite its capability to define the landslide initiation thresholds once associated with other statistical metrics like the true skill statistics and radial distance. Some research that incorporate the hydrological parameters in landslide prediction models also used the exponential or power–law function (e.g. Crozier, 1999; Monsieurs et al., 2018, 2019). Monsieurs et al. (2018) used the frequentist statistical method to define the landslide power–law threshold model line between antecedent rainfall and landslide susceptibility in western branch of the

east African rift region. Similarly, Crozier (1999) defined the exponential function between antecedent water status and daily rainfall in Wellington City, New Zealand. However, recent research (Mirus et al., 2018a; Uwihirwe et al., 2020) used the ROC curve and other statistical metrics (true skill statistics, radial distance, and threat score) to define the landslide threshold for each tested landslide predictor variable. These thresholds indicate the optimum levels in one dimension 1D of either hydrological or meteorological condition potentially linked to landslide initiation at local, regional and global scales. Her eafter,

these thresholds are therefore referred to as *single variable threshold models*. The combination of the optimum thresholds from two landslide predictor variables in two dimensions 2D as X–Y pairs is referred to as a *bilinear threshold models* firstly proposed by Mirus et al. (2018a). Some landslide studies discussed different effects that groundwater system may have on landslide initiation (Bronnimann, 2011; Cascini et al., 2010; Corominas et al., 2005; Duan et al., 2019; Hong and Wan, 2011; Trigo et al., 2005; Zhao et al., 2016). However, the asset that regional groundwater level information may have in predicting

landslide initiation on a regional scale is still underexplored. It is hypothesized that the more water stored in the catchment, the higher the probability a certain rain event will trigger landslides in a catchment. Therefore, estimates of catchment water storage could be used as a pre–event hydrological process that predispose a slope to near failure and thus be among the hydrological landslide predictor variables. However, as this information is scarce in the study area, we presuppose regional

groundwater level to be a potential proxy of the relative regional catchment storage and used as a hydrological landslide predictor variable that could be useful once incorporated in landslide threshold model definition. This research aims to include regional groundwater level information into a hydro–meteorological landslide threshold models and assess their predictive capabilities. As this type of information is not fully available, we used a parsimonious model to temporally extend regional groundwater level information to the full time period covered by the Rwanda landslide inventory. More specifically, we here tested the hypotheses that the incorporation of model derived groundwater levels in empirical landslide hazard assessment thresholds could improve the landslide warning capability in Rwanda.

## 2 Study area description

This study was conducted using data from three catchments; Kivu, upper Nyabarongo and Mukungwa (Nieuwenhuis et al., 2019); located in north western region of Rwanda, a landlocked country geographically located between 1°–3° S and 28°–31° E in central east Africa (Fig.1). The north western region is geomorphologically characterised by rounded, angular hills and headlands, mountains and volcanoes with elevation up to about 4500m and steep slope up to 55% (Fig. 2).

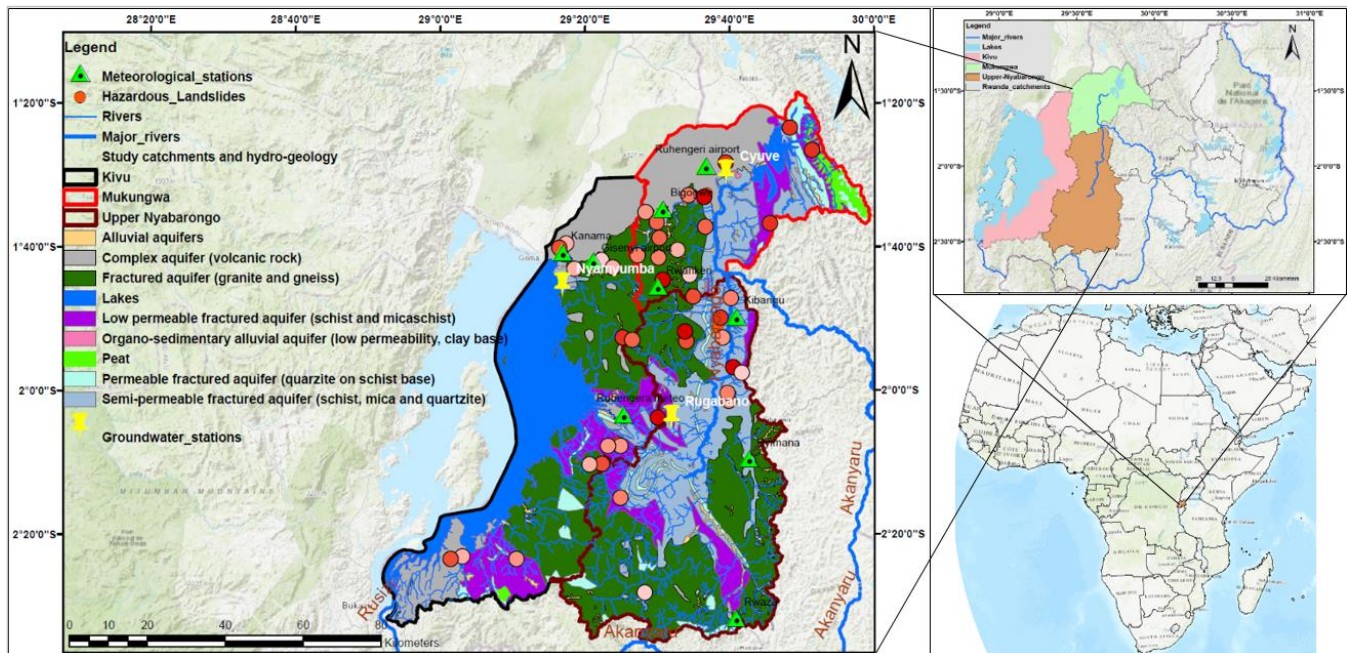

**Figure 1.** Location of the study catchments: Kivu, upper Nyabarongo and Mukungwa in Rwanda and Africa; hydro–geology of the study catchments; spatial and temporal distribution of landslides with light to dark red dots indicating old to new landslides recorded from 2006–2018 (Uwihirwe et al., 2020); groundwater stations in yellow symbols and meteorological stations in light green symbols


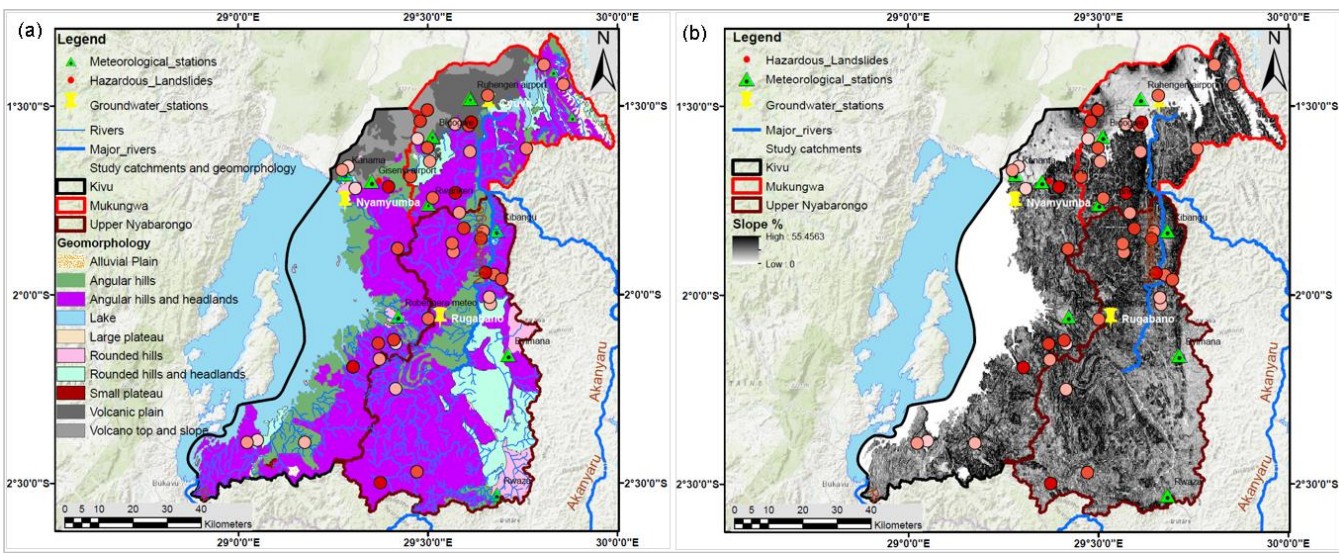

**Figure 2.** Geomorphological characteristics of the study catchments and landslides: a) landforms    b)slope; spatial and temporal distribution of landslides with light to dark red dots indicating old to new landslides recorded from 2006–2018 (Uwihirwe et al., 2020)

The total area of Kivu catchment is about 7,323 km², 2,425 km² of which is located in Rwanda. The mean annual rainfall is around 1500 mm year⁻¹ while potential evaporation is estimated at about 860 mm year⁻¹ (Fig. 3). The Kivu catchment is dominated by basaltic aquifers (volcanic rock) in the north and south west, fractured granite and gneiss aquifers in central and south east, schists and mica schists in the centre and south while pegmatite are found in intermediate areas. The upper Nyabarongo catchment is located entirely within Rwanda with an area of about 3,348 km². The mean annual rainfall is around

1200 mm year⁻¹ and potential evaporation is estimated at around 870 mm year⁻¹ (Fig. 3). Granite and gneiss aquifers are dominant in southern and to a lesser amount in north west part while quartz rich schists and mica schists dominate in central parts of the catchment (Fig. 1). The Mukungwa catchment covers a total area of 1,949 km² and is topographically dominated by the volcanic highlands region that receive abundant rainfall with a long term mean annual rainfall of around 1200 mm year⁻¹ with an estimated actual evaporation of about 800 mm year⁻¹ (Fig. 3). The hydro–geology of the catchment (Fig. 1) is

characterized by volcanic deposits with basalt in the north. Granite and pegmatite basement aquifers are found in the south western areas while quartzite and mica schist are in the south east and eastern part of the catchment. Landslides are most dominant in granite and mica schist units while basaltic units seem to be quite resistant to landslide activities as shown in Fig. 1. This can be explained by the weathering products of volcanic rocks that produce a relatively permeable top layer but tend to form a brecciated or intruded sills of low permeability layer at shallow depth and thus hampering deep groundwater recharge

and thus less prone to groundwater induced landslides. Contrarily, the weathering products of granites are generally coarse-grained that tend to develop and preserve open joint systems that increase permeability and thus fast groundwater response that leads to landslide hazards. The weathering product of mica schists include clay minerals that tend to fill up the fractures

and thus slowing the permeability. However, mica schists are prone to landslides due to rapid weathering, easy splitting along the joints and bedding planes and loss of strength induced by the high content of mica. A field based landslide inventory in the
north western region, indicated that these landslides are classified as rotational slide (34 %), flow (26 %); translational slide (17 %), fall (15 %) and complex type of mass movement (7 %) involving debris, earth and rock materials. The typical landslides are deep with estimated areal extent between $2.8\times10^1$ m$^2$ and $4.4\times10^5$ m$^2$, failure volume between $1.3\times10^1$ m$^3$ and $5.8\times10^6$ m$^3$ and mobilization rate of about 21 mm year$^{-1}$ (Uwihirwe et al., 2020).

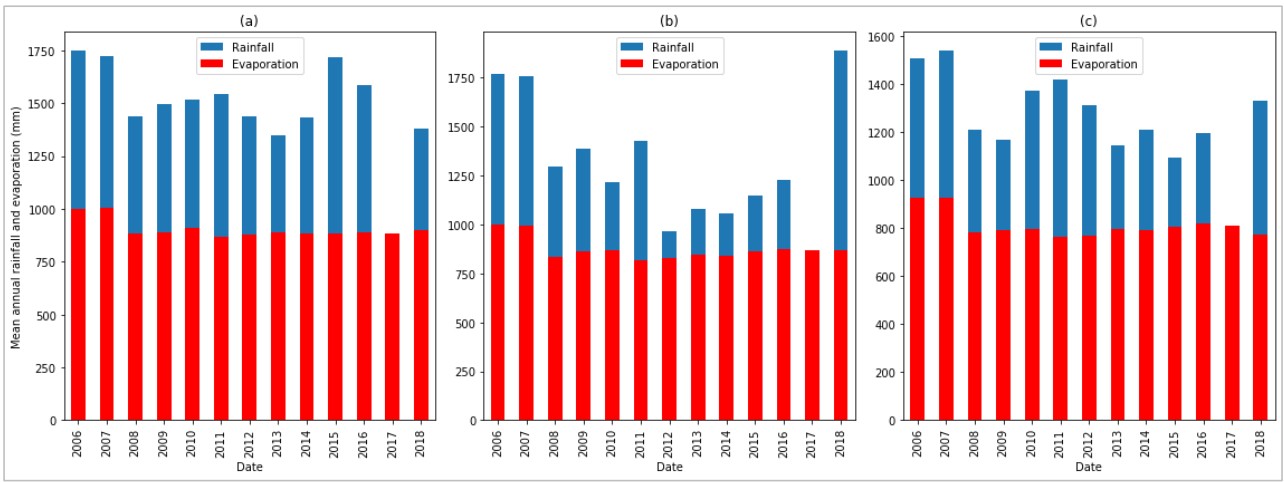

**Figure 3.** Mean catchment annual rainfall and potential evaporation in a) Kivu, b) Upper Nyabarongo and c) Mukungwa catchments

## 3 Methodology

### 3.1 Groundwater modelling: data and methodology

#### 3.1.1 Meteorological data and selection of landslide representative meteorological stations

The rainfall dataset was accessed from Rwanda meteorology agency while potential evaporation time series were calculated with Thornthwaite method (Thornthwaite, 1948) using the mean daily temperature and monthly heat index. We used time series of daily rainfall and potential evaporation from nine meteorological stations located within the studied catchments for a period of 13 years from 2006 to 2018. The meteorological stations (Fig. 1) spatially distributed in the three studied catchments were selected based on their relative proximity to the observed locations of the landslides and include Rubengera, Kanama and
Gisenyi meteorological stations in the Kivu catchment; Byimana, Kibangu and Rwaza stations in the upper Nyabarongo catchment; and Ruhengeri, Bigogwe and Rwankeri meteorological stations in the Mukungwa catchment as presented in Fig.1.

#### 3.1.2 Groundwater data and selection of landslide representative groundwater station

The time series of groundwater levels used for this study were accessed from the Rwanda water portal
(https://waterportal.rwb.rw/data/ground water). We selected three groundwater observation stations (Fig. 1) with a temporal

resolution of one day and a minimum continuous duration of one year. The three groundwater observation stations, Nyamyumba, Rugabano and Cyuve, located within the Kivu, upper Nyabarongo and Mukungwa catchments respectively, recorded data from December 2016 till December 2018. However, the intrinsic limitation of this database is linked to the coarse spatial resolution of the data recording equipment and the recorded data are insufficient to build historical time series

that match the time period of landslide inventories (2006–2018). Nevertheless, this database has been previously used for computation of water balance and catchment storage and proved to be useful in Rwanda (Nieuwenhuis et al., 2019; RWFA, 2019).

### 3.1.3 Transfer function noise (TFN) time series model

A transfer function noise (TFN) time series model describes the dynamic relationship between a single output series and one

or more input series. The TFN model was used in this research to simulate groundwater levels (model output) using both rainfall and potential evaporation as model inputs (Bakker and Schaars, 2019; Collenteur et al., 2019). With Transfer function noise modelling, the groundwater response to both rainfall and evaporation is simulated with a scaled Gamma response function. The structure of a TFN model to simulate groundwater levels is expressed with Eq. (1):

$\qquad h_t = \sum_{s=1}^{S} h_s(t) + d + r(t),$ $\qquad\qquad\qquad\qquad\qquad\qquad\qquad$ (1)

Where $h_t$ is the groundwater levels (m) at time t, $h_s(t)$ is the contribution of stresses s at time t (m d$^{-1}$), S is the total number of stresses (-) that contribute to the groundwater level change here represented by rainfall and evaporation, d is the base elevation of the model (-), and r(t) are the residuals (m). Each model can have an arbitrary number of stresses S that contribute to the head; hydrological stresses may include rainfall, evaporation, river levels, and groundwater extractions. The contribution of

stress s to the groundwater level at time t is computed through convolution with Eq. (2):

$\qquad h_s(t) = \int_{-\infty}^{t} s_s\ (\tau)\theta_s(t-\tau)d\tau,$ $\qquad\qquad\qquad\qquad\qquad\qquad\qquad$ (2)

With $s_s$ denoting the time series of stress s, and $\theta_s$ expressing the groundwater impulse response function for stress s. The groundwater response is estimated using the scaled Gamma response function that indicates the relationship between the

variation in the inputs time series (rainfall and evaporation) and the variation in the groundwater levels as in Eq. (3):

$\qquad \theta(t) = A\dfrac{t^{n-1}}{a^n \Gamma(n)}e^{-t/a},$ $\qquad\qquad\qquad\qquad\qquad\qquad\qquad$ (3)

With A denoting the scaling factor (-); a, and n are shape parameters (-) while $\Gamma$ expresses the Gamma function

### 3.1.4 Groundwater modelling approach

We used the Transfer Function Noise TFN time series Model implemented in Pastas, a new open source Python package for analysis of groundwater time series. The TFN modelling explains an observed time series (here the observed groundwater

levels) by one or more other time series (here rainfall and potential evaporation time series). The TFN model inputs time series,
rainfall and potential evaporation, were available for the entire study period 2006–2018, whereas the observed groundwater
level were available for December 2016 to December 2018. We have therefore used the two years available groundwater
observation time series and these short term data were only used for model calibration and no validation was carried out due
to the data limitations. By using the TFN modelling approach, we aimed for hindcasting and thus the reconstruction of past
groundwater levels to overlap with the time period of the recorded landslide inventory in Rwanda (2006–2018) by using the
fully available time series of rainfall and evaporation as model inputs or model stresses. Each model can have an arbitrary
number of hydrological stresses that contribute to the groundwater level changes. These hydrological stresses include rainfall,
evaporation, river levels, and groundwater extractions. For this study however, we used rainfall and potential evaporation and
assumed runoff and groundwater pumping to be negligible though not accessed in our study area. The impulse groundwater
response function to the stresses was fitted with the scaled Gamma distribution function and the calibrated parameters were A,
n, a, d as described in Sect. 3.1.3 and summarised in Appendix A. The output of the TFN model was then daily groundwater
levels $h_t$ (m) over the entire 13 years study period from 2006 to 2018. Apart from hindcasting, the TFN model spatially
extrapolated the groundwater information accounted by different precipitation and potential evaporation inputs from the nine
spatially distributed meteorological stations, Rubengera, Kanama, Gisenyi, Byimana, Kibangu, Rwaza, Ruhengeri, Bigogwe
and Rwankeri, shown in Fig. 1. The extrapolation was undertaken by changing the model inputs and model parameters at the
location of each of the meteorological stations and by implicitly relying on the main assumption here that other hydro–
geomorphological parameters do not exhibit spatial variability within the individual catchment. This is an assumption made,
given the data scarcity and some intrinsic limitation of the database in the east Africa rift region in general (Monsieurs et al.,
2018b) and Rwanda in particular. The modelled groundwater levels were standardised and used in the regional hydro–
meteorological hazard assessment threshold definition. The standardisation was computed with Eq. (4):

$$y_s = (x_i - \bar{x})/\sigma, \qquad\qquad\qquad\qquad (4)$$

Where $y_s$ is the standardised value of groundwater time series (-); $x_i$ is the value of time series (m) at time step i ; $\bar{x}$ is the
average value of time series (m); $\sigma$ is the standard deviation of time series (m); i is the subsequent time step in a time series.

### 3.2 Regional landslide assessment: data and methodology

### 3.2.1 Landslide inventory

The available landslide inventory for Rwanda contains landslides recorded from 2006 to 2018. It was accessed from the NASA
global landslide catalogue (https://data.nasa.gov/Earth-Science/Global-Landslide-Catalog/h9d8-neg4) uploaded by the
Landslide Inventory for the central section of the Western branch of the East African Rift (LIWEAR) project. The catalogue
was further extended by Uwihirwe et al. (2020) through compilation of additional rainfall induced landslides as reported from
local newspapers, blogs, technical reports and field observations. Between 2006 and 2018, the catalogue counts for 42

accurately dated landslides located within the studied region (Fig. 1). However, the detailed characteristics of these landslides such as the accurate size, types, cause and triggers are frequently not recorded by the landslide hazard reporters.

### 3.2.2 Definition of landslide hydrological and meteorological conditions

The outputs from the TFN model, groundwater levels, were used to define the landslide hydrological conditions in each of the studied catchments. The landslide hydrological conditions consist of standardized groundwater levels modelled on landslide day $h_t$ and prior to the landslide triggering event $h_{t-1}$ and were here considered as landslide cause/predisposing conditions. The meteorological conditions used here include event rainfall volumes E (mm $E^{-1}$), event rainfall intensity I (mm $d^{-1}$) as well as event duration D (d) and were considered as landslide triggers. The event duration D was defined as individual periods of days

with recorded rain interrupted by dry periods of at least two days. The event rainfall volume E was computed as the accumulated rainfall during each individual event periods of duration D. The event rainfall intensity was then computed as a ratio of E and D. Both hydrological and meteorological conditions were binary classified into landslides and no landslide conditions depending on whether they have resulted into landslide or not.

### 3.2.3 Quantification of landslide predictor variables

The landslide predictor variables which include the predisposing conditions $h_t$ and $h_{t-1}$ as well as the triggering conditions E, I and D were tested for their relevance using receiver operating characteristic (ROC) curves and the area under the curve (AUC) metrics. ROC is used as a statistical tool indicating the trade–off between false positive rate (FPR) and true positive rate (TPR) associated to each threshold level on the curve (Hong et al., 2017; Postance and Hillier, 2017; Mirus et al., 2018a; Prenner et

al., 2018). In landslide studies, the AUC is an indicator of the capacity of the test variable to correctly distinguish landslide from no landslide conditions. It is therefore used as statistical metric that compares the test variables to random guessing AUC=0.5 and thereby indicating their significance where the perfect test variable has an AUC equal to unity. The TPR and FPR corresponding to each threshold level on ROC curves are calculated with Eq. (5) and Eq. (6):

$$\text{TPR} = \frac{\text{TP}}{\text{TP+FN}}, \tag{5}$$

$$\text{FPR} = \frac{\text{FP}}{\text{FP+TN}}, \tag{6}$$

Where TP true positives or true alarms which is the number of landslides correctly predicted by the threshold model; FN false negatives or missed alarms that is the number of landslides that occurred in reality but were not predicted by the defined

threshold. FP false positives or false alarms are incorrect predictions of landslide occurrence by the threshold model while in reality there was no landslide reported. TN true negatives are correct predictions of no landslide occurrence.

### 3.2.4 Landslide thresholds definition techniques

The optimum or the most informative threshold level above which landslide are high likely to occur have been defined using two statistical techniques i.e. the maximum true skill statistic (TSS) and minimum radial distance (Rad). The true skill statistics (TSS) is expressed as a balance between the true positive rate and false positive rate as indicated in Eq. (7):

$$\text{TSS} = \text{TPR} - \text{FPR}, \tag{7}$$

Where the maximum value of $TSS$ indicates the optimum threshold for landslide initiation. For a perfect threshold model, the TSS reaches unity which indicates a zero false positive rate (FPR).

The radial distance (Rad) shows the relative distance from the defined threshold level to the perfect model or optimum point whose true positive rate (TPR) is a unit and null FPR and is computed with Eq. (8):

$$\text{Rad} = \sqrt{(\text{FPR}^2 + (\text{TPR} - 1)^2} \tag{8}$$

### 3.2.5 Single variable and bilinear threshold models and landslide predictive capabilities

According to Postance and Hillier (2017), the optimum landslide threshold model is the one that maximizes the true positive alarms (TP) while minimizing failed (FN) and false alarms (FP). Based on this criteria, the optimum threshold was here selected

among the ones defined either by maximum true skill statistics or minimum radial distance as stated in Sect. 3.2.4. These optimum thresholds were firstly plotted in 1D here referred to as *single variable threshold model line* beyond which landslide are high likely to occur. Furthermore, these optimum thresholds were combined and plotted in 2D here referred to as *bilinear threshold model line* beyond which landslide are high likely to occur. The bilinear threshold models made of hydrological and meteorological predictors were formulated using x,y pairs such as $h_t$–E, $h_t$–I, $h_{t-1}$–E and $h_{t-1}$–I and referred to as *hydro–*

*meteorological threshold models*. Furthermore, the thresholds from traditional landslide prediction models that exclusively rely on precipitation, *precipitation threshold models*, such as event–duration E–D and intensity–duration I–D were also defined in a bilinear framework and used as benchmarks for comparative performance evaluation. The predictive performance of these threshold models was evaluated using a confusion matrix and the resulting rate of positive alarms (TP), false alarms (FP), failed alarms (FN) and true negatives (TN).


## 4 Results and discussion
### 4.1 Regional groundwater modelling

The outputs of the Transfer Function Noise TFN time series model were daily groundwater levels (m) simulated over 13 years from 2006 to 2018 as presented in Fig. 4. The results demonstrate that the TFN time series model can broadly reproduce the

main features of observed groundwater level fluctuations based on the metrics of goodness of the model fit i.e. $R^2$ and RMSE between observed and simulated groundwater levels. Overall, the model explains between 60–87 % of the variance in the

observed groundwater data from the three studied catchments. The values of RMSE–groundwater levels ~ 0.09m–1.84 m similarly suggested a reasonable model fit across the catchments. More specifically, while the TFN model captures groundwater fluctuations rather well in the Kivu and Mukungwa catchments (RMSE–groundwater levels <0.5 m), the model is somewhat less robust for the upper Nyabarongo (RMSE–groundwater levels >0.5 m). The weaker model fits observed in upper Nyabarongo catchment are mostly the consequence of the relatively large distance between the groundwater well and the meteorological stations as also highlighted as potential source for poor TFN model fits by Bakker and Schaars (2019). They further postulated that TFN time series models are relatively simple, as they include only a handful number of parameters and has the higher skill to simulate groundwater levels than more detailed models.

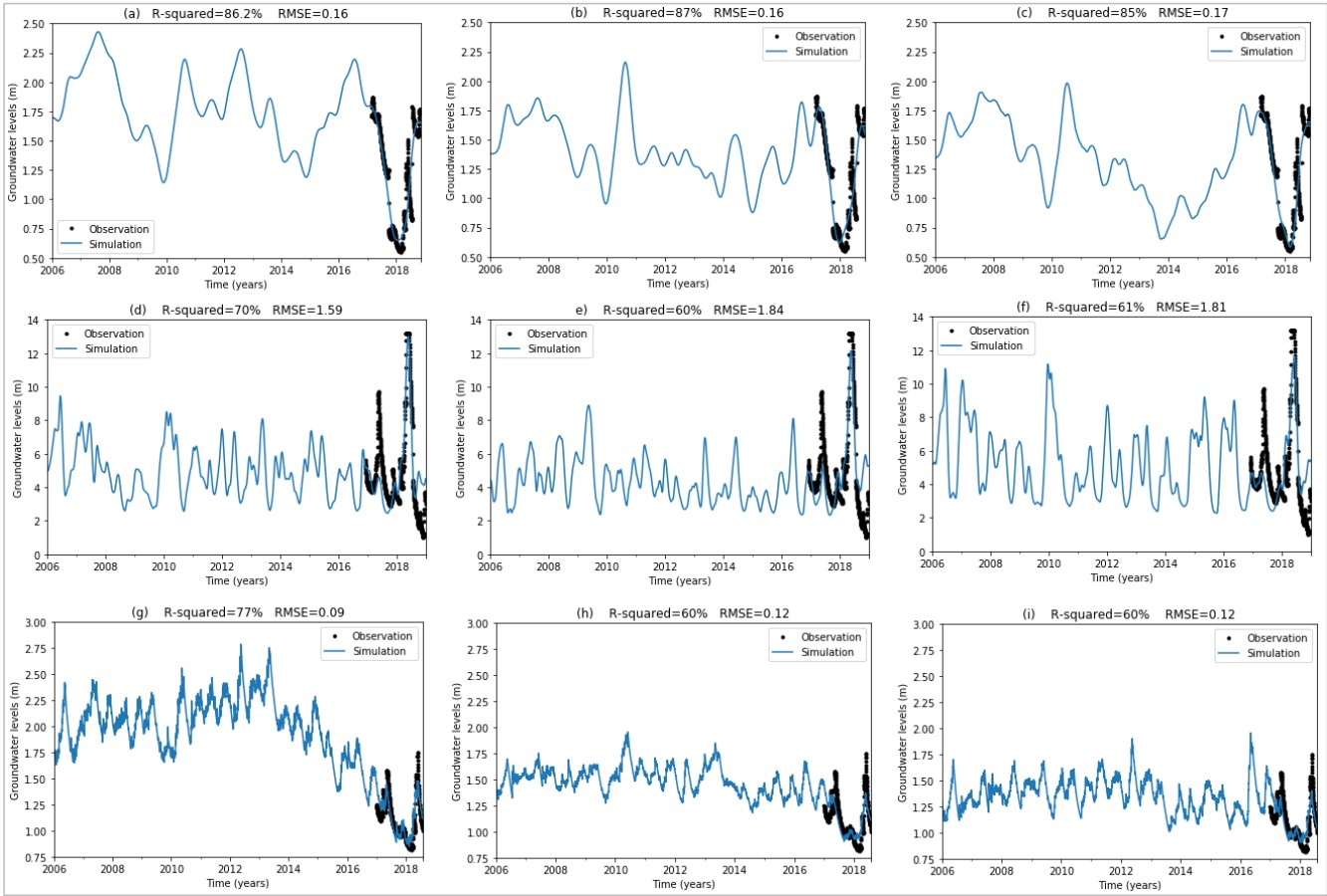

**Figure 4.** Groundwater simulation with TFN model: First row) TFN model calibrated with groundwater observations from Nyamyumba groundwater observation well; rainfall and potential evaporation Ep time series as model inputs from three meteorological stations (a) Rubengera (b) Kanama (c) Gisenyi located in Kivu catchment; Second row) TFN model calibrated with groundwater observations from Rugabano groundwater observation well; rainfall and potential evaporation Ep time series as model inputs from three meteorological stations (d) Byimana (e) Kibangu, (f) Rwaza located in upper Nyabarongo catchment; Third row)TFN model calibrated with groundwater observations from Cyuve groundwater observation well; rainfall and potential evaporation Ep time series as model inputs from three meteorological stations (g) Ruhengeri h) Bigogwe (i) Rwankeri located in Mukungwa catchment

## 4.2 Catchment standardised groundwater levels and landslides activities

The standardised daily groundwater levels and the linked landslide hazards are presented in Fig. 5 for the Kivu, upper Nyabarongo and Mukungwa catchments. The simulated groundwater levels were standardised based on the assumption that landslides occur when the groundwater levels positively deviate from the long-term mean up to a critical level for landslide initiation. The comparisons of mean daily rainfall and standardised groundwater levels across the three studied catchments, calculated by averaging of data within each catchment, indicates general similarities in terms of landslide triggering and

predisposing but also reveal systematic differences between the groundwater responses. For example, Mukungwa catchment is slowly responding and also quite drier from 2014 to 2018 than the other catchments despite its elevated landslide hazard during that period. The results indicated that landslides are likely to occur at a certain level above the long term mean groundwater level and thus justifying the importance of groundwater and catchment wetness in terms of slope failure predisposition. They also indicate that landslides occur when the catchment groundwater reaches a certain peak level above

the long-term mean which is a function of the rainfall received in the past depending on the time memory of each catchment. Even though, the most hazardous landslides in the studied catchments are shallow seated landslides which are mostly rainfall induced, the conducted field based inventory indicated that the most frequently recorded landslides in north western Rwanda are deep seated which are high likely linked to the combined effects of groundwater and other hydro–geological factors. The critical positive deviation of groundwater levels up to 3 m from the mean was noticed to be the range where most of landslide

activities happen in the studied region. However, Van Asch et al. (1999) highlighted that deep seated landslide at about 5–20 m deep are induced by rising groundwater level with about 4 m below the ground surface being the critical threshold for landslide reactivation. Hong and Wan (2011); Duan et al. (2019) forecasted the groundwater fluctuation and indicated that landslides are likely to occur when groundwater level increases by about 8 m from the datum. Even so, these absolute threshold values were not statistically approved using appropriate landslide threshold definition techniques.




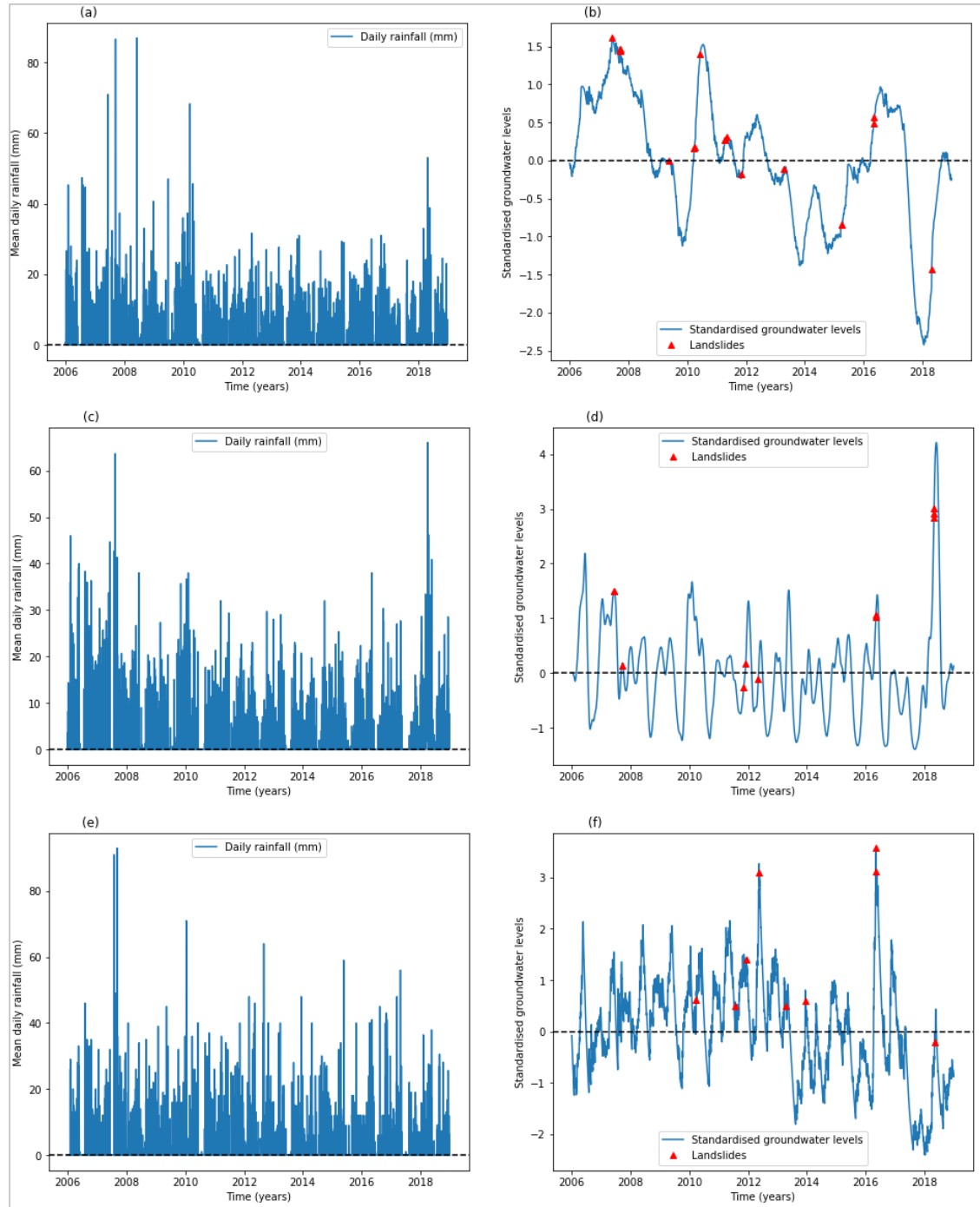

**Figure 5.** (a) Mean daily catchment rainfall and (b) catchment mean standardised groundwater simulated with TFN model using meteorological data from Kivu catchment as model inputs (c) mean daily catchment rainfall and (d) catchment mean standardised groundwater simulated with TFN model using meteorological data from upper Nyabarongo catchment as model inputs (e) mean daily catchment rainfall and (f) catchment mean standardised groundwater simulated with TFN model using meteorological data from Mukungwa catchment as model inputs; landslides represented with red dots


### 4.3 Landslide predictor variables and their discriminatory power

The discriminatory power of each landslide predictor variable was evaluated using a receiver operating characteristic (ROC) curves and area under the curve metrics as presented in Fig. 6. Based on the results, the standardized groundwater levels $h_t$ modelled on a landslide day with AUC between 0.76–0.80 and the event rainfall volume E whose AUC ranges from 0.74–0.93 were identified as the hydrological and meteorological variables with the highest discriminatory power to distinguish landslide from no-landslide conditions and thus, the most dominant control on landslide occurrence in the studied region. The

standardised groundwater levels $h_{t-1}$ recorded prior to the landslide triggering event, with AUC ranging from 0.63–0.74, were not as significant as $h_t$. This is likely a consequence of the hydro–geological properties of soil such as soil texture, presence of fissures, porosity and permeability that contribute to aquifer leakage, drainage and seepage of longer cumulated groundwater levels. Although the AUC metric was used to identify the variable with the highest skill to distinguish landslide from no-landslide conditions, it does not indicate the optimum threshold levels above which landslide are high likely to occur. Therefore

the maximum true skill statistics (TSS) and minimum radial distance (Rad) statistical metrics were used to identify the optimum thresholds represented by the dots on the ROC curves and the corresponding balance of true positive (TPR) and false positive rate (FPR) are presented in Fig. 6 and detailed in Table 1. The maximum TSS and minimum Rad indicated for example that landslides are high likely to occur when standardised groundwater levels $h_t$ positively deviate by about 0.21 to 0.48 from the long term mean and these threshold levels resulted to about 82–93 % of correct predictions of landslides i.e. true positive rate

and about 26–38 % of false positive rate. Similarly, both TSS and Rad indicated 66.8 mm event$^{-1}$ as the optimum threshold rainfall volume E with 64 % of true positive rate and 15 % of false positive rate in Kivu catchment. However, the optimum thresholds E between 44.7–63.5 mm event$^{-1}$ were defined by Rad in upper Nyabarongo and Mukungwa catchment and correctly predict about 73–92 % of landslides with 18–24 % of false positive rate. These findings indicated that the used statistical metrics TSS and Rad lead to quite similar results expressing their identical capabilities in landslide thresholds definition.



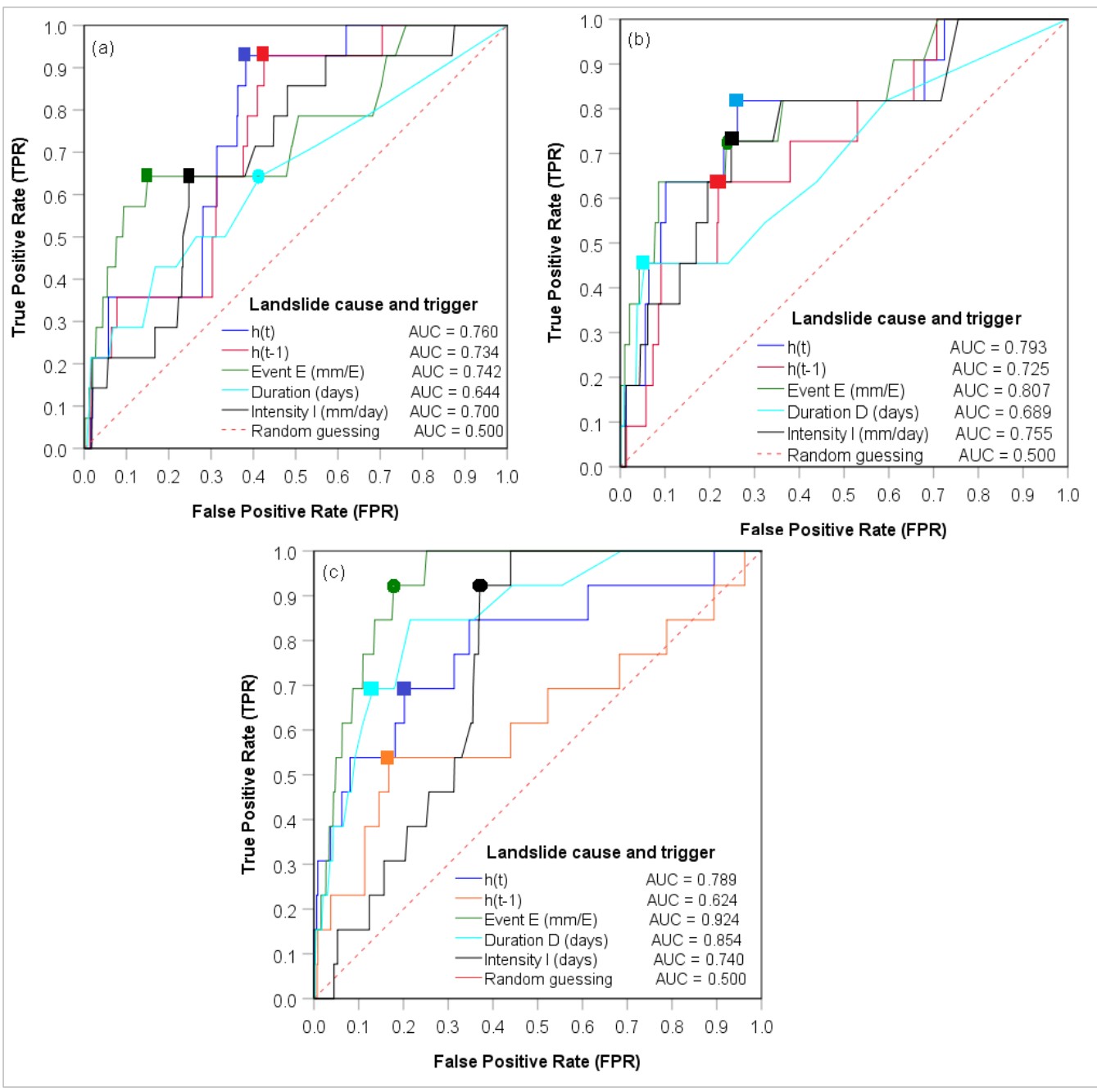

**Figure 6.** Receiver operating characteristic (ROC) curves and area under the curve (AUC) for each landslide predictor variable in the studied catchments: (a) Kivu, (b) upper Nyabarongo and (c) Mukungwa; the optimum thresholds defined using the maximum true skill statistics (TSS) are presented with square shaped markers while cycle shaped markers are thresholds defined with the minimum radial distance (Rad); once TSS and Rad reveals different threshold values the optimum (with maximum TPR and minimum FPR) is kept; once TSS and Rad reveals similar threshold values only the square shaped marker (TSS) is kept; the corresponding balance of true and false positive rate are also presented

**4.4 Comparative prediction power of single variable and bilinear thresholds models**

This research identified the landslide thresholds for each predictor variables that include the hydrological $h_t$, $h_{t-1}$ and meteorological E, I, D variables. The landslide predictive capability was evaluated for each variable in 1D here considered as single variable threshold model presented in Table 1 and by each of the blue line in Fig. 7, 8, and 9. The landslide predictive capability was also evaluated through combination of variables in 2D as X–Y pairs here considered as bilinear threshold models summarised in Table 2 and by the intersection of both blue lines in in Fig. 7, 8, and 9. A recall from Postance and Hillier (2017) indicates that the basic strategy for selection of accurate landslide threshold model is to choose the one that offers the greatest level of true positive alarms (TPR) and that provide the lowest rate of failed (FNR) and false alarms (FPR). Therefore, the findings of this research indicated that single variable threshold models either hydrological or meteorological have the greatest landslide predictive capability in terms of elevated true positive rate and low level of failed alarms as compared to the bilinear threshold models. For example with groundwater level modelled on landslide day $h_t$ with threshold values between 0.2–0.48 above the mean, 82–93 % of landslides were correctly predicted (TPR) with 25–38 % of wrongly predicted landslides (FPR). Similarly, the event rainfall intensity I between 7.5–12.5 mm d$^{-1}$ as single variable thresholds were able to correctly predict 64–92 % of landslides with 25–37 % of false alarms. Contrarily, the resulting bilinear threshold models $h_t$–I were able to correctly predict 64–85% with 8–15 % of FPR. The greatest landslide prediction capability of single variables threshold models in terms of TPR was also noticed in previously conducted research in Rwanda (Uwihirwe et al., 2020). However, it was noticed that relying on single variable threshold models that are exclusively defined using precipitation variables like event rainfall volume E, and event intensity I considered as landslide triggers could lead to biased results due to the fact that many landslides occur not only due to the trigger itself but a rather combination of both trigger and pre-event hydrological conditions. Contrarily, relying on single variable threshold models exclusively defined using hydrological variables like groundwater levels $h_t$, could lead to unbiased landslide predictions due to their high consideration of long term antecedent conditions until the day of landslide occurrence. The bilinear threshold models lead to a minimized level of false positive rate (FPR) which is the main focus behind the cause–trigger and bilinear thresholds concepts proposed by Bogaard and Greco (2018); and Mirus et al. (2018a) with a rather reduced rate of true positives (TPR).

Table 1. Single variable landslide thresholds definition with the maximum true skill statistics (TSS) and minimum radial distance (Rad) and their predictive power

| Variables | TSS threshold | TPR | FPR | FNR | TNR | TSS | RAD | RAD threshold | TPR | FPR | FNR | TNR | TSS | RAD |
|---|---|---|---|---|---|---|---|---|---|---|---|---|---|---|
| | | | | | | Kivu catchment | | | | | | | | |
| $h_t$ [a] | 0.21 | 0.93 | 0.38 | 0.07 | 0.62 | 0.55 | 0.39 | 0.21 | 0.93 | 0.38 | 0.07 | 0.62 | 0.55 | 0.39 |
| $h_{t-1}$ [b] | 0.05 | 0.93 | 0.43 | 0.07 | 0.58 | 0.50 | 0.43 | 0.05 | 0.93 | 0.43 | 0.07 | 0.58 | 0.50 | 0.43 |
| E (mm) [c] | 66.75 | 0.64 | 0.15 | 0.36 | 0.85 | 0.49 | 0.39 | 66.75 | 0.64 | 0.15 | 0.36 | 0.85 | 0.49 | 0.39 |
| D (d) [d] | 7.50 | 0.43 | 0.17 | 0.57 | 0.83 | 0.26 | 0.60 | 3.50 | 0.64 | 0.42 | 0.36 | 0.58 | 0.23 | 0.55 |
| I (mm d$^{-1}$) [e] | 10.84 | 0.64 | 0.25 | 0.36 | 0.75 | 0.40 | 0.44 | 10.84 | 0.64 | 0.25 | 0.36 | 0.75 | 0.40 | 0.44 |
| | | | | | | Upper Nyabarongo catchment | | | | | | | | |
| $h_t$ | 0.46 | 0.82 | 0.26 | 0.18 | 0.74 | 0.56 | 0.32 | 0.46 | 0.82 | 0.26 | 0.18 | 0.74 | 0.56 | 0.32 |
| $h_{t-1}$ | 0.64 | 0.64 | 0.22 | 0.36 | 0.78 | 0.42 | 0.42 | 0.64 | 0.64 | 0.22 | 0.36 | 0.78 | 0.42 | 0.42 |
| E (mm) | 90.50 | 0.64 | 0.09 | 0.36 | 0.92 | 0.55 | 0.37 | 44.70 | 0.73 | 0.24 | 0.27 | 0.76 | 0.49 | 0.36 |
| D (d) | 12.50 | 0.46 | 0.06 | 0.55 | 0.95 | 0.40 | 0.55 | 12.50 | 0.46 | 0.06 | 0.55 | 0.95 | 0.40 | 0.55 |
| I (mm d$^{-1}$) | 12.48 | 0.73 | 0.25 | 0.27 | 0.75 | 0.48 | 0.37 | 12.48 | 0.73 | 0.25 | 0.27 | 0.75 | 0.48 | 0.37 |
| | | | | | | Mukungwa catchment | | | | | | | | |
| $h_t$ | 0.48 | 0.85 | 0.35 | 0.15 | 0.65 | 0.50 | 0.38 | 0.82 | 0.69 | 0.20 | 0.31 | 0.80 | 0.49 | 0.37 |
| $h_{t-1}$ | 0.92 | 0.54 | 0.17 | 0.46 | 0.83 | 0.37 | 0.49 | 0.92 | 0.54 | 0.17 | 0.46 | 0.83 | 0.37 | 0.49 |
| E (mm) | 46.75 | 1.00 | 0.25 | 0.00 | 0.75 | 0.75 | 0.25 | 63.50 | 0.92 | 0.18 | 0.08 | 0.82 | 0.75 | 0.19 |
| D (d) | 7.50 | 0.85 | 0.22 | 0.15 | 0.79 | 0.63 | 0.26 | 7.50 | 0.85 | 0.22 | 0.15 | 0.79 | 0.63 | 0.26 |
| I (mm d$^{-1}$) | 6.78 | 1.00 | 0.44 | 0.00 | 0.56 | 0.56 | 0.44 | 7.55 | 0.92 | 0.37 | 0.08 | 0.63 | 0.55 | 0.38 |

a Groundwater levels recorded on the day of landslide   b Groundwater levels recorded prior to landslide triggering event   c Event rainfall volume
d Event duration   e Event rainfall intensity


Table 2. Landslide bilinear threshold model and warning capabilities

| Cause–Trigger | Bilinear threshold models | TPR | FPR | FNR | TNR | TSS | Rad |
|---|---|---|---|---|---|---|---|
| | | Kivu catchment | | | | | |
| $h_t$–E | $h_t>0.205$, $E>66.75$ | 0.57 | 0.07 | 0.43 | 0.93 | 0.50 | 0.43 |
| $h_t$–I | $h_t>0.205$, $I>10.84$ | 0.64 | 0.10 | 0.36 | 0.90 | 0.55 | 0.37 |
| $h_{t-1}$–E | $h_{t-1}>0.052$, $E>66.75$ | 0.57 | 0.08 | 0.43 | 0.93 | 0.50 | 0.44 |
| $h_{t-1}$–I | $h_{t-1}>0.052$, $I>10.84$ | 0.64 | 0.11 | 0.36 | 0.89 | 0.54 | 0.37 |
| E–D | $D>3.5$, $E>66.75$ | 0.57 | 0.14 | 0.43 | 0.86 | 0.43 | 0.45 |
| I–D | $D>3.5$, $I>10.84$ | 0.36 | 0.06 | 0.64 | 0.94 | 0.29 | 0.65 |
| | | Nyabarongo catchment | | | | | |
| $h_t$–E | $h_t>0.457$, $E>44.7$ | 0.73 | 0.08 | 0.27 | 0.92 | 0.64 | 0.29 |
| $h_t$–I | $h_t>0.457$, $I>12.48$ | 0.73 | 0.08 | 0.27 | 0.92 | 0.65 | 0.28 |
| $h_{t-1}$–E | $h_{t-1}>0.636$, $E>44.7$ | 0.55 | 0.07 | 0.45 | 0.93 | 0.48 | 0.46 |
| $h_{t-1}$–I | $h_{t-1}>0.635$, $I>12.48$ | 0.64 | 0.07 | 0.36 | 0.93 | 0.56 | 0.37 |
| E–D | $D>12.5$, $E>44.7$ | 0.45 | 0.05 | 0.55 | 0.95 | 0.40 | 0.55 |
| I–D | $D>12.5$, $I>12.48$ | 0.36 | 0.01 | 0.64 | 0.99 | 0.36 | 0.64 |
| | | Mukungwa catchment | | | | | |
| $h_t$–E | $h_t>0.483$, $E>63.5$ | 0.77 | 0.11 | 0.23 | 0.90 | 0.66 | 0.25 |
| $h_t$–I | $h_t>0.483$, $I>7.55$ | 0.85 | 0.15 | 0.15 | 0.85 | 0.70 | 0.21 |
| $h_{t-1}$–E | $h_{t-1}>0.921$, $E>63.5$ | 0.46 | 0.03 | 0.54 | 0.97 | 0.43 | 0.54 |
| $h_{t-1}$–I | $h_{t-1}>0.921$, $I>7.55$ | 0.54 | 0.06 | 0.46 | 0.94 | 0.48 | 0.47 |
| E–D | $D>7.5$, $E>63.5$ | 0.85 | 0.14 | 0.15 | 0.86 | 0.71 | 0.21 |
| I–D | $D>7.5$, $I>7.55$ | 0.77 | 0.06 | 0.23 | 0.94 | 0.71 | 0.24 |

## 4.5 Comparative analysis of the warning capabilities of landslide hydro–meteorological thresholds and precipitation based thresholds

The landslide hydro–meteorological threshold models defined as X–Y pairs in a 2D bilinear framework and their warning capabilities in Kivu catchment are presented in Fig. 7. The combined groundwater level–event rainfall intensity $h_t$–I [$h_t$>0.205, I>10.84 mm d$^{-1}$] threshold model outperforms other combinations in terms of true positive alarms with about 64 %. Comparing the predictive capabilities of $h_t$–I, a hydro–meteorological threshold model, to I–D, a precipitation threshold model, significant improvement of about 28 % in terms of the rate of true alarms was obtained from $h_t$–I as compared to I–D. This confirms the

high landslide prediction and warning capability of hydro–meteorological thresholds over precipitation based thresholds. However, there was no significant improvement from E–D to $h_t$–E and $h_{t-1}$–E in terms of true alarms. This suggests that the combinations involving event rainfall volume E have lower landslide warning skill than the ones that consider the event rainfall intensity I. This may be explained by the fact that rainfall event volume E is estimated over various time scale D making E an unstandardized variable which could be normalized by the respective time duration and thus, favouring the event rainfall

intensity I. Unexpectedly, there was no significant improvement in terms of reduced false alarms FPR by the tested landslide hydro–meteorological threshold models as compared to the precipitation based threshold models in Kivu catchment.

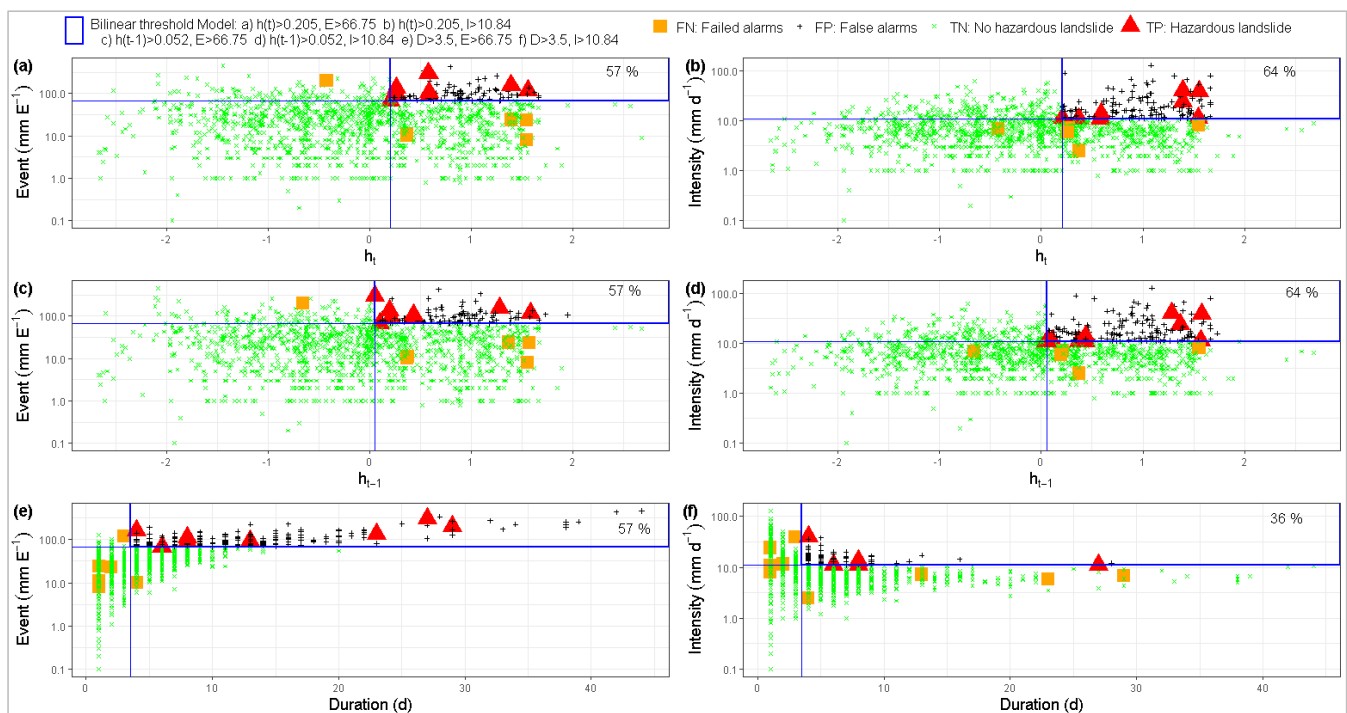

**Figure 7.** Landslide warning capabilities of the hydro–meteorological and precipitation threshold models: .(a) $h_t$–E; (b) $h_t$–I;
(c) $h_{t-1}$–E; (d) $h_t$–E; (e) E–D; (f) I–D in Kivu catchment

The defined landslide hydro–meteorological threshold models in upper Nyabarongo catchments are presented in Fig. 8. Similar to Kivu catchment, the landslide hydro-meteorological threshold models $h_t$–E, $h_t$–I, $h_{t-1}$–E and $h_{t-1}$–I performs much higher with 55–73 % of correctly predicted landslides (TP) than precipitation threshold models E–D and I–D with around 36–45 % of true alarms. A significant reduction of the rate of failed /missed alarms (FN) with about 37 % from I–D to $h_t$–I and about 28 % from E–D to $h_t$–E was also observed. Unexpectedly, there was no significant improvement in terms of reduced false alarms by the landslide hydro–meteorological thresholds as compared to the landslide precipitation thresholds.

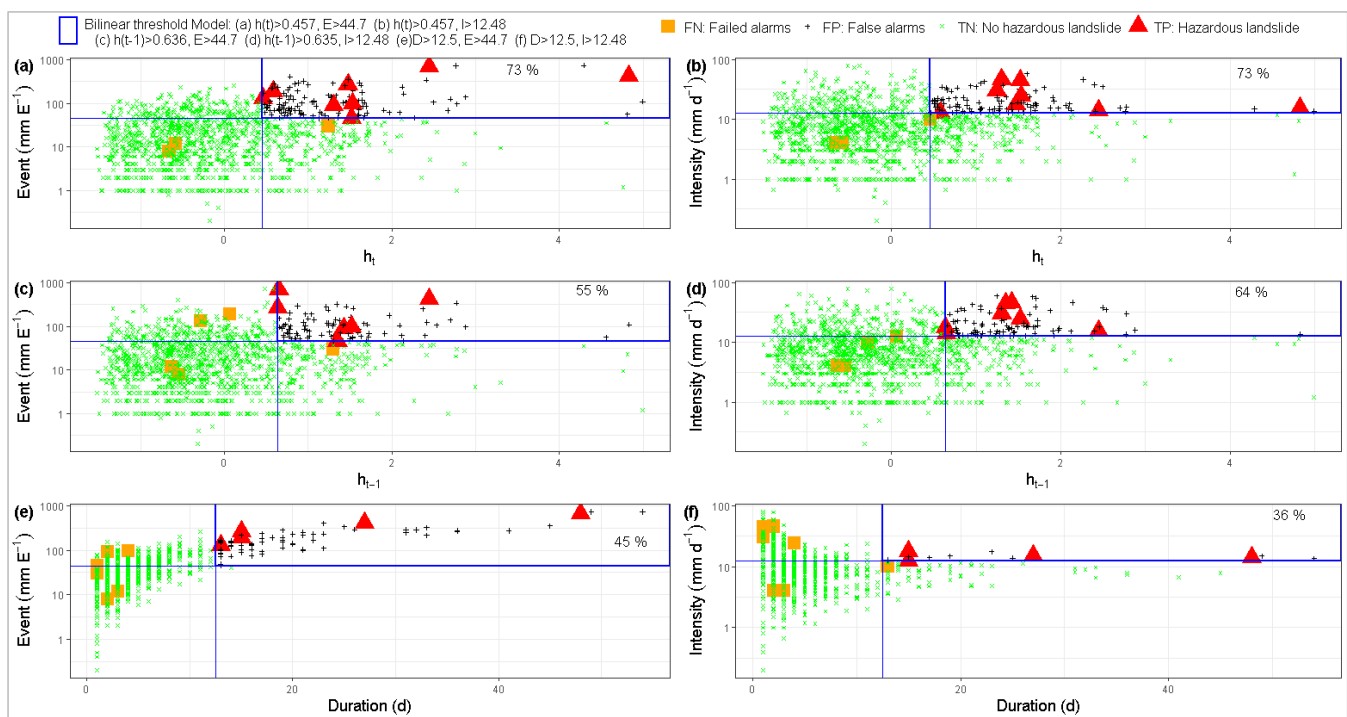

**Figure 8.** Landslide warning capabilities of the hydro–meteorological and precipitation threshold models: .(a) $h_t$–E;  b) $h_t$–I;  (c) $h_{t-1}$–E;  (d) $h_t$–E;  (e) E–D;  (f) I–D in upper Nyabarongo catchment

The defined landslide hydro–meteorological threshold models in Mukungwa catchment are shown in Fig. 9. Although, there was no significant improvement in terms of false positive alarms (FP) reduction as expected, the best landslide hydro–metrological thresholds models $h_t$–I outperforms the precipitation based threshold I–D models in terms of elevated rate of true positive alarms TP with about 85 % as compared to 77 % and low rate of failed alarms FN with 15 % compared to 23 %. The highest prediction level in terms of true alarms with 85 % was observed from both $h_t$–I and E–D hydro–meteorological and precipitation based threshold models. Contrary to Kivu and upper Nyabarongo catchments, precipitation based threshold models E–D and I–D performed quite similar to $h_t$–I and even better than other tested hydro–meteorological threshold models in Mukungwa catchments. This could be explained by the catchment specific hydro–geological characteristics that probably

makes the catchment to be a very slow groundwater responding system and thus, a rather more precipitation induced landslide than groundwater levels.

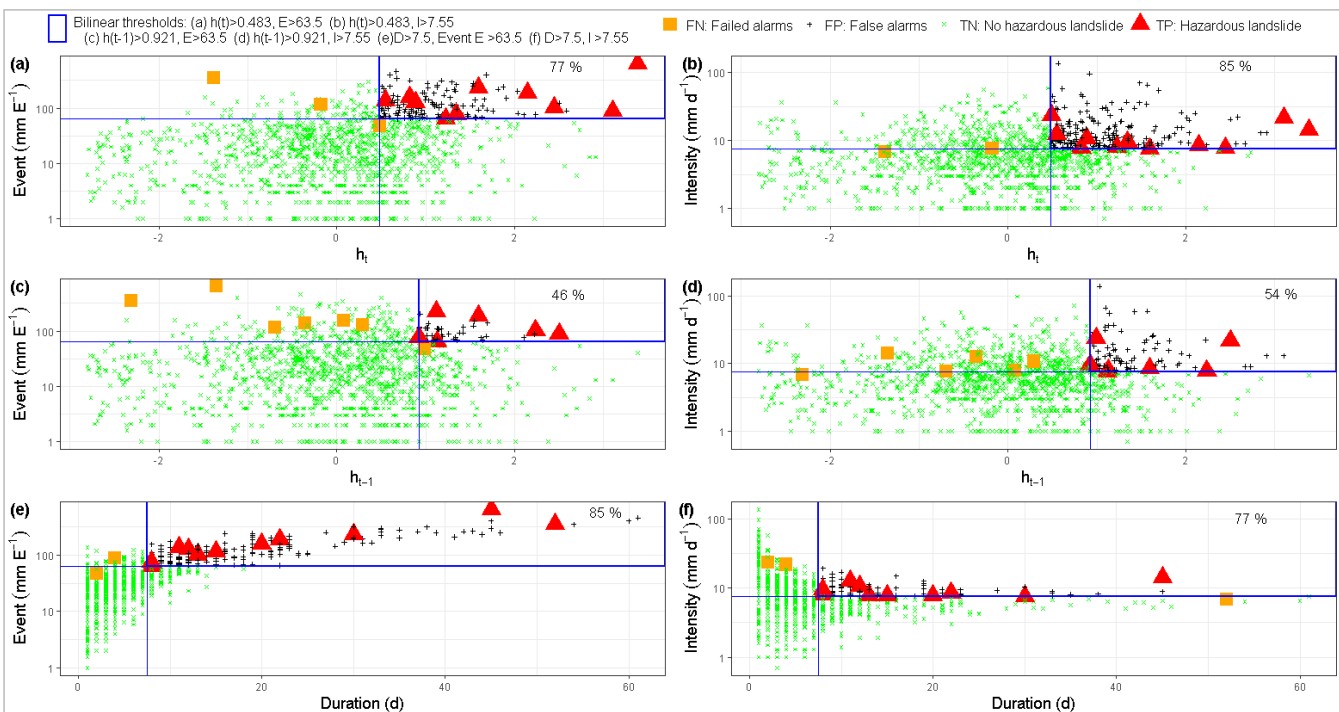

**Figure 9.** Landslide warning capabilities of the hydro–meteorological and precipitation threshold: (a) $h_t$–E; (b) $h_t$–I; (c) $h_{t-1}$–
E; (d) $h_t$–E; (e) E–D; (f)I–D in Mukungwa catchment

## 4.6 Adaptability and limitation of the defined landslide threshold models

Within the framework of this research, we defined the landslide empirical hydro–meteorological thresholds using continuous historical precipitations time series and groundwater level time series as proxy for the catchment water storage. We mainly
analysed the difference in landslide thresholds and warning capabilities as a result of the differences in catchment water storage, estimated from the groundwater responses to precipitation. It was observed that the catchment with complex or slow groundwater responding system such as Mukungwa, the warning capability of the groundwater based thresholds have less performance as compared to the fast and clear groundwater responding systems like Nyabarongo and Kivu catchments. This is truly owed by the catchment specific hydrogeological and geomorphological characteristics. Nevertheless, the in deep
analysis of the hydrogeological and geomorphological differences between the three catchments and how they could be among the explanatory factors of the observed difference in landslide thresholds and the warning capabilities was not fully conducted. However, with reference to Fig.1 and Appendix B, Mukungwa catchment is hydrogeologically characterized by complex aquifer in volcanic rocks and thus being a complex or slow groundwater responding system. This is probably due to the

weathering products of volcanic rocks that produce a relatively permeable top layer but tend to form a brecciated or intruded sills of low permeability layer at shallow depth and thus hampering deep groundwater recharge. Contrarily, Nyabarongo and Kivu catchments are dominated by fractured granites with overall high transmissivity and recharge and hence fast and clear groundwater responding systems (Appendix B). The weathering products of granites are generally coarse grained that tend to develop and preserve open joint systems that increase permeability and thus fast groundwater response. In Nyabarongo and Kivu catchments therefore, the landslide warning capability of groundwater based thresholds performed higher than precipitation thresholds as opposed to Mukungwa catchment. This is to say that in regions with very slow groundwater responding system where rainfall induced shallow landslides prevail, precipitation based thresholds can still practically be useful for landslide prediction and warning. However, the need for hydrological thresholds is true for both shallow and deep seated landslides (Cascini et al., 2010; Corominas et al., 2005; Duan et al., 2019; Hong and Wan, 2011) and thus, being more powerful than precipitation based thresholds. More studies also confirm the high warning capability of hydro–meteorological thresholds over precipitation based thresholds after incorporation of either soil moisture or catchment storage (Ciavolella et al., 2016; Mirus et al., 2018a; Prenner et al., 2018; Thomas et al., 2019; Wicki et al., 2020). According to Uwihirwe et al. (2020), a study conducted in Rwanda to define precipitation thresholds, the highest predictive capability of precipitation based threshold in a bilinear framework that used the antecedent precipitation API and event rainfall intensity I as $API_{30}$–I, was about 68 % of true alarms associated with 27 % of false alarms. However, this prediction level was further improved through this research by considering the catchment specific groundwater levels where the best predictor $h_t$–I was able to correctly predict 85 % of landslides (TP) with 15 % of false alarms. Although, the catchment water storage would have been a better landslide predictor, this type of information is scarce. Therefore, the groundwater level was considered as a proxy and used as a hydrological landslide predictor variable in our research. The component of groundwater has been on one hand considered as landslide triggering factor and on the other hand as landslide predisposing factor (Cascini et al., 2010; Corominas et al., 2005; Duan et al., 2019; Hong and Wan, 2011). Being a hydrological parameter, it was subjectively considered as landslide predisposing factor and plotted on x axis of a 2D plot as a cause in a cause–trigger framework. However, the neutral use of groundwater levels (neither trigger nor cause) in a single variable threshold model $h_t$ provided excellent prediction results up to 93 % of correctly predicted landslide and only 7 % of failed alarms with a rather high rate of false alarms up to 38 %. The adopted approach for hydro–meteorological and/or bilinear threshold model definition aimed to reduce the rate of false alarms associated with single variable thresholds and follows the cause–trigger concept (Bogaard and Greco, 2018) in which the groundwater levels as cause were combined with precipitation variables as trigger in a bilinear framework (Mirus et al. 2018a). We have tested different combinations of the optimum hydrological and meteorological threshold variables such as $h_t$–E, $h_t$–I, $h_{t-1}$–E, and $h_{t-1}$–I and the combination of groundwater levels on the day of landslide and event rainfall intensity $h_t$–I proved to have higher skill for landslide prediction and warning with high rate of true alarms 64–85 % and reduced rate of false alarms 8–15 % as compared to other combinations. We remain convinced that the combination of appropriate threshold variables into cause–trigger framework should consider the time scale of each variable to avoid overlapping time scales between hydrological and meteorological variables. However, the combinations of $h_t$–E, and $h_t$–I may led to overlapping time scale between

groundwater levels and rainfall event. This would be very true for longer time scale triggers and very fast groundwater responding system with very short time memory which was not the case in our studied catchments. To account on this constraints, we have also considered the groundwater level recorded prior to landslide triggering events $h_{t-1}$–E and $h_{t-1}$–I combinations but the result was not as significant as $h_t$–E and $h_t$–I. In this research, the single variable and bilinear threshold models were adopted rather than power law models commonly used in landslide precipitation threshold like intensity–duration and event–duration. These single variable and bilinear threshold models were selected based on our dataset that displays most of the landslide conditions in the upper right corner of the plots as shown in Fig. 7, 8, 9 and the achieved landslide predictive capabilities summarized in Table 1 and Table 2. Although one is free to choose any other model that fit the dataset, the single variable and bilinear threshold models proved to be more efficient for hydro–meteorological threshold model definition (Mirus et al., 2018a; Uwihirwe et al., 2020). Furthermore, the transfer function noise TFN time series model was used for groundwater modelling because of its simplicity, less data requirement and above all its higher skill in groundwater simulation (Bakker and Schaars, 2019; Collenteur et al., 2019). However, like other models, 100 % of the observed data cannot fit the model. Therefore, the modelled groundwater data used to define the hydro–meteorological threshold may be prone to minor errors. Additionally, the spatial extrapolation of groundwater information relied on the main assumption that other hydro–geomorphological parameters do not exhibit spatial variability within the individual catchment. This is an assumption made, given the data scarcity in the east Africa rift region in general (Monsieurs et al., 2018b) and Rwanda in particular. Lastly, the landslide inventory used for this study relied largely on the information from government reports, newspapers, and other media where many landslide events are likely to be missed. Although, the reliance on these data sources is likely to lead to a bias towards larger landslide events and those with impact to society, this landslide inventory is the most comprehensive currently available in the study area.

## 5 Conclusion

This research aimed to improve the landslide forecast quality by incorporating the catchment specific groundwater levels as a proxy for regional water storage. A parsimonious transfer function noise (TFN) time series model was used to simulate the groundwater levels that temporally match with the available landslide inventory. Based on the statistical measures of goodness of fit, the root mean square error (RMSE–groundwater levels <0.5 m) and the explained variance ($R^2$ >60 %), the TFN time series model demonstrates sufficient skill to simulate groundwater levels. The standardized groundwater levels $h_t$ modelled on a landslide day with AUC between 0.76–0.80 and the event rainfall volume E whose AUC ranges from 0.74–0.93 were identified as the hydrological and meteorological variables with the highest discriminatory power to distinguish landslide from no landslide conditions and thus, the most dominant control on landslide occurrence in the studied region. The single variable threshold model derived from groundwater levels $h_t$ indicated the highest landslide prediction and/or warning capability with about 85–93 % of true positive alarms despite the resulting rate of false alarms between 26–38 %. Similarly, the single variable threshold models derived from precipitation intensity I and volume E reveal also high landslide predictive skill in terms of true positive alarms with about 64–100 % associated with 15–44 % of false alarms. However, it was noticed that relying on single

variable threshold models exclusively derived from precipitation variables like E and I considered as landslide triggers could lead to biased results due to the fact that many landslides occur not only due to the trigger itself but a rather combination of both trigger and pre-event hydrological conditions. Contrarily, relying on single variable threshold models exclusively defined using hydrological variables like groundwater $h_t$, lead to unbiased landslide predictions due to their high consideration of long-term antecedent conditions until the day of landslide occurrence. Further combination of the optimum groundwater and precipitation thresholds as bilinear threshold models reduced the rate of false alarms by about 18–28 % at the expense of reduced rate of true positive alarms by about 9–29 % and thus being less advantageous than single variable threshold models. However, the hydro-meteorological threshold models defined in bilinear framework as $h_t$–I indicated higher landslide predictive skill in terms of true positive alarms (64–85 %) than traditional threshold model I–D (36–77 %) that exclusively rely on precipitation. Furthermore, the integration of catchment specific groundwater levels in landslide hazard assessment in Rwanda improved the landslide prediction and warning capabilities of the existed precipitation based threshold that used the antecedent precipitation API as a proxy for hydrological condition and event intensity I as a meteorological condition. Overall, the incorporation of observed and model derived groundwater variables in an empirical statistical approach and the use of regional specific hydrological characteristics improve the landslide prediction capacity as compared to the exclusive use of global precipitation based threshold models.

**Data availability.** The datasets for this research can be accessed at https://doi.org/10.4121/15040446.v1

**Competing interests.** The authors declare that they have no conflict of interest

**Author contributions.** JU collected data, implemented the modelling approaches, conducted statistical analysis, conceptualise and prepared the manuscript storyline. MS corrected the manuscript storyline, shaped the discussion and contributed to the perfection of the manuscript. TAB proposed the modelling approaches, verified the model outputs, shaped the manuscript storyline and contributed to the perfection of the manuscript.

**Acknowledgements.** This research was undertaken as part of PhD studies sponsored by the Strengthening Education for Agricultural Development (SEAD) project in Rwanda and is extremely appreciated. We are highly grateful to the Schlumberger Foundation Faculty for the Future for the partial scholarship. We are so thankful to the Landslide Inventory for the central section of the Western branch of the East African Rift (LIWEAR) project for sharing the landslide inventory. We are thankful to the Rwanda meteorological agency and Water Resource Board for offering access to the meteorological and hydrological datasets used for this research.

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

**Appendices**

**Appendix A**

Table A1. Final values of the calibrated parameters

| Parameters | Kivu catchment | | | Upper Nyabarongo catchment | | | Mukungwa catchment | | |
|---|---|---|---|---|---|---|---|---|---|
| | Rubengera | Kanama | Gisenyi | Byimana | Kibangu | Rwaza | Ruhengeri | Bigogwe | Rwankeri |
| A | 0.75 | 0.40 | 0.63 | 0.84 | 0.68 | 0.82 | 0.97 | 0.31 | 0.20 |
| a | 81.64 | 3.88 | 117.69 | 10.54 | 13.19 | 8.97 | 1000 | 257.23 | 128.23 |
| n | 3.45 | 5.63 | 2.34 | 4.92 | 3.78 | 5.64 | 0.79 | 0.92 | 0.91 |
| d | 1.11 | 0.092 | -0.91 | 2.42 | 5.66 | 6.27 | 0.48 | 1.61 | 1.43 |

With A denoting the scaling factor (-); a and n are shape parameters (-) while d is the base elevation of the model (-) as described in Sect. 3.1.3.






**Appendix B**

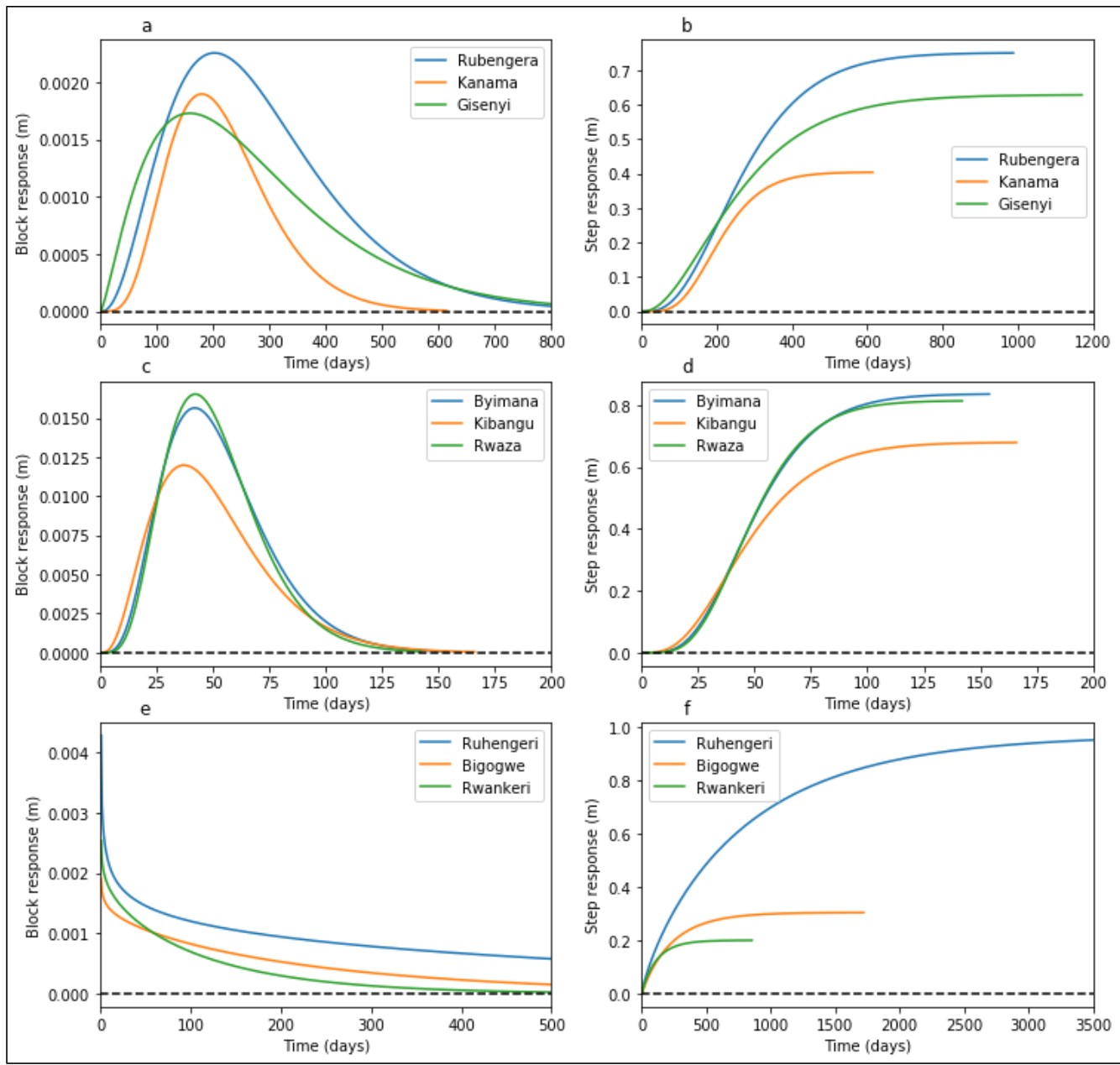

**Figure B1**. Groundwater response function a) Block response and b) Step response to both rainfall and potential evaporation recorded from three landslide representative meteorological station in Kivu catchment c) Block response and d) Step response to both rainfall and potential evaporation recorded from three landslide representative meteorological station in Upper Nyabarongo catchment e) Block response and f) Step response to both rainfall and potential evaporation recorded from three landslide representative meteorological station in Mukungwa catchment