# Peer review of "Integration of observed and model derived groundwater levels in landslide threshold models in Rwanda"

_Natural Hazards and Earth System Sciences, 2021_

## Author Response (AR1)

**Responses to comments_RC1**

Dear Prof. Valentino,

We thank you for the detailed comments and constructive feedback. Please find below, the replies to the comments and approaches to address the comments and corrections in the revised manuscript :

**Comment 1**: On page 7, rows 184-186, the Authors state that "the extrapolation was undertaken by keeping the parameters constant in each of the three catchments and by implicitly relying on the main assumption here that the subsurface characteristics do not exhibit spatial variability within the individual catchment." This basic hypothesis appears rather strong and oversimplified, as it seems not to take into account the morphology of the area under study and the high variability of altitude and slope. It is believed that the extreme orographic variability of the area can have a significant influence on the estimation of the groundwater level. The model adopted has the merit of being simplified, but in this territorial context there is a risk of making completely misleading estimates. Moreover, taking the same data from a single groundwater station and extending this data not only to the sites of the 3 rain stations (for each sector) but to the whole sector seems rather risky

**Response:** We agree that using the same data from a single groundwater station and extending this data to the sites of the three rain stations and to the locations of landslides in the entire catchment through modelling approaches is risky. However, as you also mentioned, this study was conducted in a data scarce region with very sparse groundwater monitoring wells and ground-based rain stations. We therefore understand the adopted data driven approach as a step forward in data scarce areas that can however be further improved depending on future availability of data with fine spatial resolution.

We have paraphrased the paragraph in lines 184–186 for more clarification. In fact, we did consider the spatial variability of groundwater levels as affected by changes in rainfall and evaporation (model inputs) recorded at each of the three rain stations while assuming other hydro-geomorphological parameters to be constant within each individual catchment. However, we are aware that the fine spatial resolution of the groundwater monitoring wells and the rain stations would have led to more improved results in the context of high orographic and morphologic variability territory like Rwanda. We have added a detailed discussion of these limitations in Sect. 4.6 of the revised manuscript.

**Comment 2**: The second concern deals with the types of landslides considered. On page 7, row 200, the Authors declare that they use a catalogue that includes "42 accurately dated landslides located within the studied region". However, the main characteristics of these 42 landslides are not reported and not explained. The study area is intensely affected by several types of landslides: rainfall-induced landslides, deep seated landslides and also rock falls. In addition, in this area landslides occur very often on steep road cut-slopes, where the influence of the water table as a predisposing or triggering cause remains to be proven. Authors are invited to incorporate some comments related to these shortcomings.

**Response:** As mentioned on page 20, line 483–485, the landslide inventory used for this study relied largely on the information from government technical reports, newspapers, and other media where the accurate information about landslide characteristics such as sizes, types,

triggers and causes is frequently not reported by the landslide hazards reporters. Currently, this landslide inventory is the most available with the highest temporal accuracy (landslide occurrence day) in the study area. However, based on literature and field observations, general information about landslide characteristics (size and type) that prevail in the study area have been added in Sect. 2 of the revised manuscript. In addition, we have provided a short discussion of the potential effects of this limitation in Sect. 4.6 of the revised manuscript.

**TECHNICAL CORRECTIONS**

**Comment 3**: Page 10, row 281: "Ruhengeri" instead of "Ruhengeli".

**Response**: Corrected accordingly

**Comment 4**: Section 4.2 and Figure 4: the correlation between landslide triggering and increase of groundwater level is not so evident.

**Response**: The Section 4.2 and Figure 4 intend to show that landslides are likely to occur when groundwater level increases at a certain level above the long-term mean as a results of the rainfall received in the past despite some few exceptional landslides that may have been probably induced exclusively by rainfall. In our opinion, the correlation (strong linear correlation) between landslide triggering conditions (here rainfall) and increase in groundwater levels is not supposed to be evident due the time lag between groundwater response and rainfall as affected by the time memory of each catchment. This is also linked to the hydro-geotechnical properties of soil like hydraulic conductivity, permeability and soil texture that contribute to subsequent interplay between infiltration, evaporation and drainage and thus the change in groundwater levels.

**Comment 5**: Figure 5: the caption refers to square shaped markers for TSS and cycle shaped marker for Rad, but cycle shaped markers are only reported in Figure 5.c: is it correct or some markers are missing?

**Response:** The markers on Figure 5 are correct. We preferred to keep only the square shaped marker (TSS) on the curve once both true skill statistic TSS and Radial distance Rad reveal similar threshold values (Table 1). Once TSS and Rad reveals different threshold values, the optimum (with maximum TPR and minimum FPR) is kept. We have added this information to the caption of Fig. 5 for a better clarification.

**Comment 6:** Page 20, row 473: "...but the result was not as significant as..." instead of "...but the results was not as significant as...".

**Response**: Corrected accordingly

**Responses to comments_RC2**

95   Dear Dr. Giulio Castelli,

We are grateful for your overall positive feedback on the manuscript and important suggestions, corrections and comments. Below our responses and ways adopted to address the
100   raised issues:

**Comment 1:** Lines 176-177. The choice of calibrating the model with the later years (instead of the earlier ones) is rather uncommon. Why is this so? Was a proper validation carried out, besides the calibration? What software/methodology was used for the calibration? Which parameters were calibrated?

105   **Response:** We used the Transfer Function Noise TFN time series Model that was implemented in Pastas, a new open source Python package for analysis of groundwater time series. The TFN modelling explains an observed time series (here the observed groundwater levels) by one or more other time series (here rainfall and evaporation time series). The TFN model inputs time series, rainfall and evaporation, were available for the entire study period 2006–2018, whereas
110   the observed groundwater level were available for December 2016 to December 2018. We have therefore used the two years available groundwater observation time series and these short term data were only used for model calibration and no validation was carried out due to the data limitations. By using the TFN modelling approach, we aim to hindcast the groundwater levels and get a full time series covering the entire period of the landslide inventory in Rwanda (2006-
115   2018) by using the fully available time series of rainfall and evaporation as model inputs also called model stresses. Each model can have an arbitrary number of hydrological stresses that contribute to the groundwater level changes. These hydrological stresses include rainfall, evaporation, river levels, and groundwater extractions. For our research however, we used the rainfall and potential evaporation and assumed runoff and groundwater pumping to be
120   negligible though not accessed in our study area. The impulse groundwater response function to the stresses was fitted with the scaled Gamma distribution function. The calibrated parameters (Eq. (3)) were A, n, a, d with A denoting the scaling factor (-); a, and n are shape parameters (-) and d is a constant or base elevation of the model as summarized in section 3.1.3. We have made some edits to the M&M section to improve Sect. 3.1.4 of the revised manuscript.

125   **Comment 2.** Lines 184-187. With reference to comment 1 in RC1, I am not fully convinced of the answer given by the Authors. Please state in the M&M section that this is an assumption made given the data scarcity in the area, and provide a justification of the choice, eventually citing suitable references.

**Response:** We agree with your suggestion to add in the M&M section the information that the
130   assumption was made given the data scarcity in the east African Rift region in general and Rwanda in particular. In the revised manuscript we have added this information and provided additional references (e.g: Monsieurs et al., (2018)).

**Comment 3**: Lines 195-200: With reference to comment 2, RC 1, I have to say that even here authors should declare that the database has some intrinsic limitations in the M&M section.
135   Kindly cite some papers using the same database to show some example of its usage.

**Response**: Yes it is true that the used database have some intrinsic limitation and we have added such information in the methodological part and some examples (References) of the previous database use have been provided accordingly (e.g: Nieuwenhuis et al. (2019; Rwanda Water and Forestry Authority (2017)).

140 **Comment 4**: Line 260: It would be useful to understand which is the relative RMSE value, e.g. for example RMSE/mean_groundwater_depth. Moreover, it is not fully clear which was the final value of the calibrated parameter.

**Response**: The suggestion to add the relative RMSE value is a good one and we agree that it will make the RMSE more meaningful once added for example the RMSE–groundwater level

145 where the relative value here is the groundwater level. We have made a summarised Table indicating the final values of the calibrated parameters and is appended to the revised manuscript (Appendix A).

**Comment 5**: Paragraph 4.5: Can the differences in the three watersheds in terms of warning capabilities and thresholds be explained by their geo-morphological differences? How this is

150 related to the comment at line 184–187?

**Response:** Within the framework of this research study, we defined the landslide empirical hydro–meteorological thresholds using continuous historical precipitations time series and groundwater level time series as proxy for the catchment water storage. We mainly analysed the difference in landslide thresholds and warning capabilities as a result of the differences in

155 catchment water storage, estimated from the groundwater responses to precipitation. It was observed that the catchment with complex or slow groundwater responding system such as Mukungwa, the warning capability of the groundwater based thresholds have less performance as compared to the fast and clear groundwater responding systems like Nyabarongo and Kivu catchments. This is truly owed by the catchment specific hydrogeological and

160 geomorphological characteristics. Nevertheless, the in deep analysis of the hydrogeological and geomorphological differences between the three catchments and how they could be among the explanatory factors of the observed difference in landslide thresholds and the warning capabilities was not fully conducted. However, with reference to Fig. 1 and Appendix B, Mukungwa catchment is hydrogeologically characterized by complex aquifer in volcanic rocks

165 and thus being a complex or slow groundwater responding system. This is probably due to the weathering products of volcanic rocks that produce a relatively permeable top layer but tend to form a brecciated or intruded sills of low permeability layer at shallow depth and thus hampering deep groundwater recharge. Contrarily, Nyabarongo and Kivu catchments are dominated by fractured granites with overall high transmissivity and recharge and hence fast

170 and clear groundwater responding systems (Appendix B). The weathering products of granites are generally coarse grained that tend to develop and preserve open joint systems that increase permeability and thus fast groundwater response. In Nyabarongo and Kivu catchments therefore, the landslide warning capability of groundwater based thresholds performed higher than precipitation thresholds as opposed to Mukungwa catchment.

175 A point of discussion about these possible effects of the hydrogeology on the observed differences in landslide thresholds and warning capabilities have been provided in Sect. 4.6 of the revised manuscript. Appendix B showing the groundwater response curves for each of the three study catchments has been added to the revised manuscript. In the study area section (Sect. 2), the general information about the catchment typical geomorphological characteristics

180 such as landforms and slope have been provided in addition to the hydrogeology.